# Viroporins of Mpox Virus

**DOI:** 10.3390/ijms241813828

**Published:** 2023-09-07

**Authors:** Kingshuk Basu, Miriam Krugliak, Isaiah T. Arkin

**Affiliations:** Department of Biological Chemistry, The Alexander Silberman Institute of Life Sciences, The Hebrew University of Jerusalem, Edmond J. Safra Campus, Jerusalem 91904, Israel; kingshuk.baso@mail.huji.ac.il (K.B.); miriamkru@savion.huji.ac.il (M.K.)

**Keywords:** Mpox virus, viral ion channels, ion channel assay, molecular dynamic simulation

## Abstract

Mpox or monkeypox virus (MPXV) belongs to the subclass of Poxviridae and has emerged recently as a global threat. With a limited number of anti-viral drugs available for this new virus species, it is challenging to thwart the illness it begets. Therefore, characterizing new drug targets in the virus may prove advantageous to curbing the disease. Since channels as a family are excellent drug targets, we have sought to identify viral ion channels for this virus, which are instrumental in formulating channel-blocking anti-viral drugs. Bioinformatics analyses yielded eight transmembranous proteins smaller or equal to 100 amino acids in length. Subsequently, three independent bacteria-based assays have pointed to five of the eight proteins that exhibit ion channel activity. Finally, we propose a tentative structure of four ion channels from their primary amino acid sequences, employing AlphaFold2 and molecular dynamic simulation methods. These results may represent the first steps in characterizing MPXV viroporins en route to developing blockers that inhibit their function.

## 1. Introduction

Among the recent outbreaks of different viral strains, Mpox or monkeypox is a potentially alarming disease. The scientific community is trying to gain insight into the epidemiology, dynamics of transmission and molecular characteristics of the virus responsible for the infection. Mpox virus, or MPXV, the etiological agent of Mpox, is a member of Poxviridae that emerged in 1958 among primates, and since 1970, it has been found to infect humans [1]. Though the disease was initially contained primarily within central and West African countries, after May 2022, several cases were found in European countries [2,3,4]. The symptoms of Mpox are similar but milder than smallpox, e.g., muscle aches, headache, fever, rash, and lymph node swelling [5], which is of no surprise since the coding regions of MPXV and variola virus, the etiological agent of smallpox, are over 96% identical [6]. Finally, Mpox is a zoonotic disease, but its animal reservoir is yet to be uncovered [7].

Despite a large number of studies, only two FDA-approved drugs are available in the market against Mpox: Tecovirimat [8,9] and Brincidofovir [10,11]. Both of these drugs are known to inhibit viral replication. Tecovirimat is known to target the VP37 protein, which assists in extracellular viral particle formation [9]. On the other hand, Brincidofovir targets the DNA polymerase enzyme, which aids viral DNA synthesis [11].

In addition to the above protein targets, viroporins as a family represent another attractive viral component to inhibit [12,13,14,15]. For example, the M2 $H^{+}$ channel of the Influenza A virus serves as the target of aminoadamantanes [16] that have been known to inhibit influenza since 1964 [17]. Consequently, research has been devoted to identifying and targeting viral ion channels, such as JC virus Agnoprotein [18,19,20], to increase the scope of anti-viral drugs [21,22,23,24].

MPXV is an oval-shaped virus with a length of 220–450 nm [25], larger than HIV or SARS-CoV-2 viruses [26]. Similarly, the double-stranded DNA genome size of MPXV of 197 kb [27] is significantly larger than the aforementioned two pathogens and is thought to encode around 200 proteins [28]. However, to the best of our knowledge, viroporins have not been reported for the virus.

In the current study, we have employed a bacteria-based characterization of MPXV transmembrane proteins and analyzed their potential to function as ion channels. Following our experimental investigation, we made tentative structural predictions for the top-performing ion channels based on AlphaFold2 [29] and molecular dynamic simulations. These biochemical and structural characterizations can cast light on how to formulate future anti-viral drugs against this disease.

## 2. Results

### 2.1. Sequence Analysis

Ion channel proteins transverse through the lipid bilayer and, as such, are transmembrane in nature. Accordingly, we searched the 200 kb genome of MPXV [6] for proteins containing transmembrane segments with Phobius [30] and TMMHM [31]. Due to the large number of proteins identified and the overall small size of viroporins, we arbitrarily decided to limit our study to proteins equal to or smaller than 100 amino acids in length. With this criterion in hand, eight small transmembrane domains were chosen: C20.5L, A15.5L, gp124, gp063, gp066, gp125, gp081, and gp120. Their corresponding accession numbers, lengths, and probable TM domains are listed in Table 1, and their sequences are given in Appendix A.

### 2.2. Experimental Channel Examination

#### 2.2.1. Expression and Membrane Incorporation of Proteins

As expounded below, we expressed the above-stated proteins in three bacterial systems suitable for investigating channel activity. These systems were used to analyze numerous viroporins and, as such, demonstrated their applicability and suitability [21,22,23,24,32,33,34].

#### 2.2.2. Negative Assay

In the negative assay, a viral protein is expressed at increasing levels, and if it exhibits channel activity, it causes commensurate permeabilization of the bacterial membrane. In other words, growth retardation is observed as a phenotypical expression of the channel [33]. This method is particularly helpful for checking the activity of viral channel blockers since they alleviate the growth impairment [21,22,23,24,32,33].

Figure 1 shows the growth rates of bacteria expressing the different aforementioned viral proteins. The growth was measured as a function of OD_600nm_ with respect to time, from which a maximal growth rate was determined. Upon increasing the concentration of the IPTG inducer, a corresponding reduction in growth rate is observed for the bacteria that express most chimeras, albeit at different levels. Specifically, at 60 µM IPTG, bacteria that express gp120, C20.5L, gp081, A15.5L, gp063, gp125, gp066, and gp124, exhibited a relative growth reduction of: 31%, 38%, 42%, 54%, 75%, 77%, 97%, and 103%, respectively.

#### 2.2.3. Positive Assay

In the positive assay, a $K^{+}$-uptake deficient strain of bacteria is used in the analysis. The bacteria cannot grow in LB media unless it is supplemented by [$K^{+}$] or expresses a channel capable of $K^{+}$ transport [35]. Hence, growth enhancement is observed as a phenotypical expression of the viral protein [36], which is reciprocal to the negative assay discussed above. Note that in this instance, the expression level of the channel is lower than in the negative assay to prevent deleterious excess membrane permeabilization [37].

Figure 2 depicts the effect of the expression of different viral proteins on cellular growth. Upon expression of gp120, A15.5L, gp081, C20.5L, and gp063 proteins, we observed a 208%, 221%, 101%, 82%, and 47% increase in bacterial growth in comparison to their uninduced forms, respectively. On the other hand, upon induction of gp125, gp066, and gp124, no significant change in bacterial growth rate was observed.

#### 2.2.4. pH Assay

In the final experimental assay, the $H^{+}$ intake kinetics of the bacterial cells were checked upon expression of the viral proteins. In this assay [34], a particular strain of bacteria expresses pHluorin, a pH-sensitive green fluorescent protein [38]. pHluorin has two excitation maxima at 390 nm and 460 nm, and the ratio between them is a function of the pH within the cell. Therefore, the activity of a protein capable of $H^{+}$ transport can be detected by fluorescence upon injection of an acidic solution into the media.

Figure 3 shows the pattern of pH change in the bacterial cells upon expression of different proteins. Bacteria that express the gp124, A15.5L, gp063, gp125, gp081, and gp066 chimera exhibit a pronounced pH drop indicative of their $H^{+}$ conductivity, while those expressing C20.5L and gp120 do not. The slopes of the linear regressions of the data are listed in Figure 4. Here, it is worth mentioning that the concentration of IPTG chosen for the different proteins was not uniform. We only studied the maximal pH influx for a particular channel. Furthermore, another intermediate concentration of IPTG was selected to show an increment of pH influx upon increasing IPTG concentration.

### 2.3. Computational Studies

The results of the negative assay (Figure 1), positive assay (Figure 2), and fluorescent-based $H^{+}$ influx assay (Figure 3) are summarized as a 3D plot in Figure 4 and in tabulated form in Appendix A. Based on their effectivity in each assay, ranks were assigned, and based on averaged rank, five proteins that may serve as viroporins were selected for further computational studies: A15.5L, gp081, gp120, gp063, and C20.5L. We, therefore, proceeded to employ molecular modeling to obtain a primary insight into the ion channels’ structural features. In brief, AlphaFold2 structural prediction yielded models for the proteins, followed by molecular dynamics simulation analyses of their structural integrity within a lipid bilayer.

#### 2.3.1. Structure Prediction

After the success of AlphaFold [29], AlphaFold2 (AF2 from Google CoLab [39]) has emerged as a more efficient tool for making structural predictions of proteins from their amino acid sequences. Moreover, AF2 has been found to predict the structures of transmembrane proteins with higher accuracy than soluble proteins [40].

A key consideration in the modeling process was the oligomeric nature of the proteins. Due to their small size and channel activity, the proteins are most likely oligomeric, in line with other viroporins [41]. However, since the oligomeric number is unknown, we used AF2 to generate structures of differing oligomeric structures ranging from trimer to hexamer for each of the putative viroporins. Subsequently, two criteria were used to select the probable oligomeric number: (i) the presence of a pore throughout the structure since it is necessary for conduction, and (ii) the highest level of sequence-wise confidence score marked by the pLDDT score [42].

Appendix A shows the per-residue confidence estimates (pLDDT plots) obtained from AF2 for the different oligomeric structures. It can be concluded from the distribution of confidence levels that all protein sequences, except A15.5L, show the highest confidence level for their corresponding trimeric structure. A15.5L, on the other hand, shows the highest confidence level for a tetrameric structure. Finally, we did not obtain structures containing a lumen surrounded by a helical bundle for either oligomer of gp081.

Appendix A shows the top structures (rank one) chosen for further computational studies, obtained from AlphaFold2 with corresponding position-wise confidence value plots.

#### 2.3.2. Molecular Dynamics Simulations

The structural models obtained from AF2 were subsequently analyzed by molecular dynamics simulations. The probable transmembrane region of each oligomer was inserted in a hydrated lipid bilayer and subjected to a 100 ns trajectory. The stability of the proteins was examined by measuring the RMSD values of the backbone for each simulation with respect to time, as shown in Figure 5. It is evident from the plots that every structure converges to stable conformers with relatively similar RMSD values at the higher time scale of each trajectory. Such stable structures with similar RMSD values for each protein were overlapped (approximately 70 structures for each protein) to yield a representative structure of the proteins within the lipid environment, as shown in Figure 6.

## 3. Discussion

The Mpox virus is an emerging threat to the healthcare sector, and options for medicinal remedies are wanting. Channels as a family are excellent drug targets, and by inference, viral ion channel blockers represent an attractive anti-viral strategy to inhibit viral infectivity. However, to achieve such an aim, we first need to identify potential viral ion channels in MPXV and characterize their activity. To achieve this goal, we employed three bacteria-based channel assays to evaluate the activity of putatively predicted ion channels from the corresponding protein sequences.

Ion channels are transmembranous in nature. Therefore, it is necessary first to identify proteins with one or more transmembrane domains to explore channels. From different sequence analysis techniques, we found the following MPXV proteins which have probable transmembrane domain(s): C20.5L, A15.5L, gp124, gp063, gp066, gp125, gp081, and gp120 (see Table 1 and Appendix A, which includes the transmembrane domain lengths and positions). We studied the channel activity of these proteins using three different bacteria-based biochemical assays and proposed the probable structure of the effective ion channels using computational tools.

Bacterial channel assays are based on three different techniques used to quantify the activity of ion channels expressed in the organism’s inner membrane. A phenotypic difference in the bacteria is obtained upon channel expression at varying levels in all three assays. Finally, reproducible membrane targeting and incorporation is achieved by fusing the viral proteins to the maltose binding protein [21,22,23,24,32,33,34].

The negative assay entails retarding the growth of the host bacteria upon increasing expression of viral ion channels due to membrane permeabilization (under the control of Isopropyl-β-d-1-thiogalactopyranoside). We found a significant setback in bacterial growth upon increasing the expression of all the aforementioned proteins except for the gp124 chimera (Figure 1).

The positive assay does the opposite job of the negative assay. In a $K^{+}$-uptake deficient bacterial system, ion channel expression causes an increase in growth rate [35]. Interestingly, in this assay, we found pronounced ion channel activity effects for gp120, A15.5L, gp081, gp063, and C20.5L chimeras (Figure 2).

The third assay we employed measured quantitatively the kinetics of proton influx upon induction of the ion channels. For the expression of gp124, A15.5L, gp063, gp125, gp081, and gp066-based chimeras, we found a pronounced drop in pH, whereas, in the case of C20.5L and gp120, such $H^{+}$ conduction was insignificant.

Based on the different quantitative channel activity assessments of the eight proteins, we ranked each between 1 and 8. Different color markings shown in Figure 4 describe the relative activity of the ion channel activity of the chimera. Based on the ranking for each chimera in the three bacterial assays, we calculated an average ranking for each system, and as an order of activity, they can be written in descending order as: A15.5L > gp081 > gp063 = gp120 > C20.5L > gp124 = gp125 > gp066. Finally, among these eight proteins, only the top five exhibited activity in the positive and negative bacterial assays. Therefore, these viral proteins can be considered potential channels worthy of further investigation by computational tools.

While the sequences of the above putative channels are known, alongside their activity in bacteria-based assay, their structural details are unavailable to us. Therefore, AlphaFold2-based structural predictions [29,39] were conducted to yield the proteins’ putative structures. Since the proteins mostly likely form oligomers when functioning as channels, oligomers were constructed using AlphaFold2. Upon using both selection criteria (see Section 2.3.1), one structure was selected from each protein from several probable oligomeric structures depending upon the confidence value of the TM region (see Appendix A). Finally, we did not obtain a suitable ion channel forming pore helical assembly for gp081.

The probable structures of the four ion channel proteins are depicted in Appendix A. A central hollow core with concentric assembled α-helices was found for all the structures. It is worth mentioning that AlphaFold2 is very effective in predicting homo-oligomeric assembly [39]. Therefore, the assembled structural outputs, without lipid environments, show trimeric or tetrameric assembly with an arrangement of polar-hydrophobic-polar zones (Appendix A), which is suitable for placing them within a triphasic lipid environment.

For C20.5L, gp063, and gp120, simple trimeric models were chosen (Appendix A). For C20.5L, the segment-wise confidence level is more or less uniform in nature (Appendix A). In the case of gp063, as evident from Appendix A, the confidence level of the less structured, charged, hydrophilic segments (with cyan protein segment) have relatively lower confidence, but the hydrophobic segments have much higher confidence (higher than 70%). On the other hand, for gp120, the overall level of confidence is lower. The probable TM segment has a value slightly higher than 50%.

Interestingly, the A15.5L structure exhibits different properties. The sequence of the monomeric protein segment has a discontinuous TM zone. From segments 25 to 29, there are two lysines and one aspartic acid residues. In the predicted AlphaFold2 structure, this hydrophilic segment has formed a turn, making two hydrophobic TM domains closer to each other, making a “V” shaped motif. From Appendix A, it can be seen that, upon oligomer formation, the arrangement has aligned it in a way so that the turn makes a charged end in the polar-hydrophobic-polar pattern. This makes the model suitable to fit within a lipid environment.

MD simulations were conducted to examine whether AlphaFold2 predicted if proteins are stable within the membrane environment. To this end, structures obtained from AlphaFold2 were embedded within a model 1,2-dipalmitoyl-sn-glycero-3-phosphocholine membrane for molecular dynamic simulation studies.

All four structures exhibit stability during the course of 100 ns within the lipid bilayer environment. Figure 5 shows the change in RMSD values for the backbone of each protein with respect to time. In all cases, the system converges during the course of the simulation. Finally, Figure 6 shows the structures of the four primary models of C20.5L, A15.5L, gp063, and gp120 placed within a lipid bilayer.

## 4. Materials and Methods

### 4.1. Searching Transmembrane Proteins

The genome of the MPVX virus (GenBank: ON563414.3) was analyzed for the presence of potential viroporins by searching every open reading frame for transmembrane helices using a hidden Markov model with TMHMM [43,44]. The same sequences were cross-checked with Phobius, another web-based model that checks transmembrane topology and signal peptide sequences [30,44]. Subsequently, all proteins with one or more predicted transmembrane helices were ranked according to size from smallest to largest, and the proteins smaller than 100 amino acids were selected for investigation: C20.5L, A15.5L, gp124, gp063, gp066, gp125, gp081 and gp120.

### 4.2. Bacterial Expression

For the channel assays, we expressed the genes of the aforementioned proteins within three types of *Escherichia coli* strains, namely, DH10B, LB650, and LR1. The DH10B strain was purchased from Invitrogen (Carlsbad, CA, USA). LB650, a $K^{+}$-uptake deficient strain [35], was a kind gift from Professor K. Jung (Ludwig-Maximilians Universität München) and Professor G.A. Berkowitz (University of Connecticut). The LR1 strain for measuring $H^{+}$ flux [34] was a kind gift from Professor M. Willemoës and Professor K. Lindorff-Larsen (Københavns Universitet).

### 4.3. Plasmids

All the above-listed proteins were expressed in the corresponding cell lines as fusion proteins with the maltose binding protein employing the pMal-p2X vector (New England Biolabs; Ipswich, MA, USA). The sequences were synthesized by GenScript (Piscataway, NJ, USA). In all cases, expressions were triggered by Isopropyl-β-d-1-thiogalactopyranoside (IPTG).

### 4.4. Chemicals

IPTG was purchased from Biochemika-Fluka (Buchs, Switzerland). All other chemicals were purchased from Sigma-Aldrich laboratories (Rehovot, Israel).

### 4.5. Growth Media

Lysogeny Broth (LB) was used for all bacterial growth unless noted otherwise. LBK was similar to LB except that KCl replaces NaCl at 10 gr/L. All media contained ampicillin at 50 µg/mL.

### 4.6. Negative Ion Channel Assay

DH10B bacteria containing the desired plasmids were grown overnight, after which they were diluted 50 times and grown until the OD_600nm_ reached a value of 0.2. From the culture, 50 μL were added to 96-well plates containing 50 μL of other required solutions. Induction of the desired proteins was achieved by adding IPTG at different concentrations. The bacterial growth kinetics were measured by monitoring the OD_600nm_ of the resultant solutions using an infinite M200 pro (Tecan Group; Männedorf, Switzerland) or LogPhase 600 from BioTek (Santa Clara, CA, USA). The incubation time for each experiment was 16 h, and the experimental temperature was 37 °C with a shaking speed of 700 rpm. Every 15 min, data points were collected and plotted as a function of time.

### 4.7. Positive Ion Channel Assay

A positive assay was conducted employing the same protocol as the negative ion channel assay listed above, except that LB50 bacteria, a $K^{+}$-uptake deficient strain, was used [35]. Moreover, LBK media was used with 150 mM KCl instead of NaCl for overnight growth.

### 4.8. Acidity Assay

This assay relies on the fluorescence of a pH-sensitive green fluorescent protein [38] present within the bacterial chromosome [34]. For this experiment, overnight bacterial cultures were diluted in a 1:50 ratio in LB media and further grown until the OD_600nm_ reached 0.1. Proteins were expressed using IPTG at different concentrations. After one hour of induction, each of the cell growths was brought to an equivalent concentration of 0.2 OD_600nm_ and was centrifuged at 2058× *g* for 5 min. The pelleted cells were suspended in McILvaine Buffer [45] (200 mM $Na_{2}HPO_{4}$ and 0.9% NaCl adjusted to pH 7.6 with 0.1 M citric acid). Subsequently, 200 μL of this suspension was added to 96-well plates (Nunclon f96 Microwell Black Polystyrene, Thermo Fisher Scientific; Waltham, MA, USA), whereby each well contained 30 μL of McILvaine Buffer. In the plate, three wells contained McILvaine buffer and three bacterial cultures without induction as control. The fluorescence data were measured with a microplate reader (Infinite F200 Pro, Tecan Group; Männedorf, Switzerland) at ambient temperature. Two fluorescence measurements were taken at each time point: 520 nm emission and either 390 nm or 466 nm excitation. At the starting point, 70 μL of 300 mM citric acid was added to the bacteria culture, and fluorescence intensity data was measured for 120 s for each wavelength.

### 4.9. Computational Studies

Primary protein sequences (oligomeric) were inputted into AlphaFold2.ipynb using ColabFold v1.5.2 to obtain tentative three-dimensional structures [39]. The outcome structures were subsequently embedded within a pre-equilibrated 1,2-dipalmitoyl-sn-glycero-3-phosphocholine bilayer downloaded from the website of D. Peter Tieleman [46] such that the protein helical axes remain perpendicular to the membrane plane. The bilayer had 128 lipid molecules and 3783 water molecules. After initial structural alignment, the clashing lipids and water molecules within the range of 2 Å were removed using InflateGro methodology developed by Tieleman and co-workers [47]. Position-restraints (100,000 kcal mol${}^{-1}$ Å${}^{-2}$) on protein-heavy atoms were used to ensure that the protein does not change during energy minimization accomplished using the steepest descent minimization algorithm with a tolerance of 500 kj mol${}^{-1}$ nm${}^{-1}$.

MD simulations for all the systems were conducted for a duration of 100 ns using Gromacs version 2022.3 [48,49,50,51,52] using an extended version of GROMOS96 53A6 force field [53]. The LINCS algorithm with an integration time step of 2 fs was used in all cases to constrain the length and angles of hydrogen atoms [54]. Atomic coordinates were saved at every 500 ps. Reference temperatures were set at 323 K, and solvent, lipids, and proteins were coupled separately to a Nose-Hoover temperature bath [55,56] with a coupling constant value τ=0.5 ps. Pressure coupling was obtained with a Parrinello-Rahman barostat with τ=2 ps [57,58]. A 1.2 nm cut-off was set for van der Waals interactions. The neighbor list was updated every 10 fs. Electrostatic parameters were calculated using fourth-order Particle Mesh Ewald (PME) long-range electrostatics [59], and cut-off short-range electrostatics were set to 1.2 nm.

The simulation box contained 125 lipid molecules and 3700 water molecules (the number varies as the size of the box changed for larger proteins), approximately with ${Na}^{+}$ and ${Cl}^{-}$ counter ions. Water molecules were fitted in the FLEXSPC model [60], and the total number of atoms was in the 17,000–25,000 range, depending on the protein sequence and oligomerization state. RMSD values with respect to time were extracted, and structures of the simulated proteins were visualized and analyzed with VMD software version 1.9.4a51 [61].

## 5. Conclusions

Potential Mpox viral ion channels were studied to assist future anti-viral drug discovery. Both experimental and computational studies were performed to characterize the activity of ion channels and to provide a preliminary idea of their tentative structural models. From three types of bacteria-based biochemical assays (negative, positive, and pH-flux), an average performance ranking was made, and out of eight proteins, five effective channels were obtained. These five sequences were inputted into AlphaFold2, an AI structure prediction approach, to determine each protein’s tentative structure. These structures also showed good stability within the triphasic lipid environment upon MD simulation, and from an overlay of the converged structure, we proposed preliminary structures of four transmembrane ion channels of the Mpox virus. Finally, this study can potentially facilitate future progress in drug design against the Mpox virus.

## Figures and Tables

**Figure 1 ijms-24-13828-f001:**
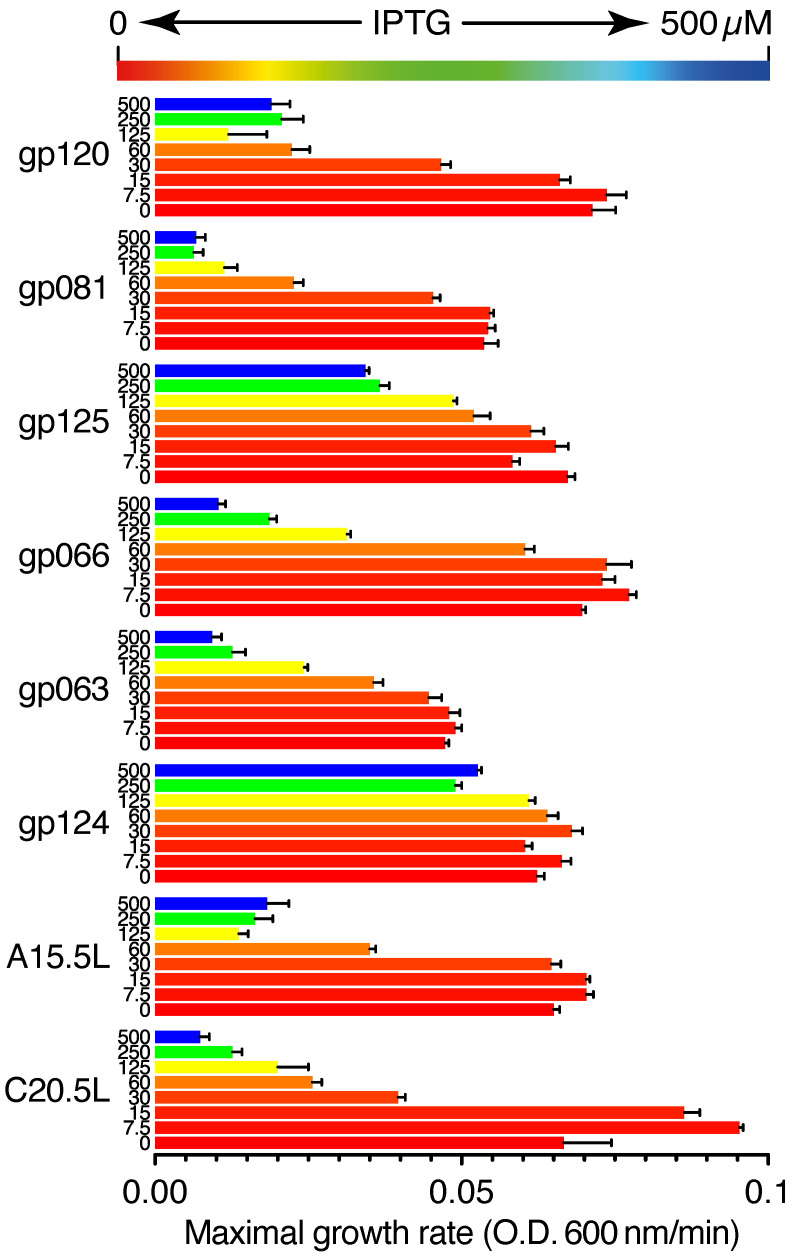
Negative genetic test for viroporin activity. The impact of expression of viral protein chimera upon bacterial growth, as a function of the concentration of protein inducer IPTG from 0 to 500 µM.

**Figure 2 ijms-24-13828-f002:**
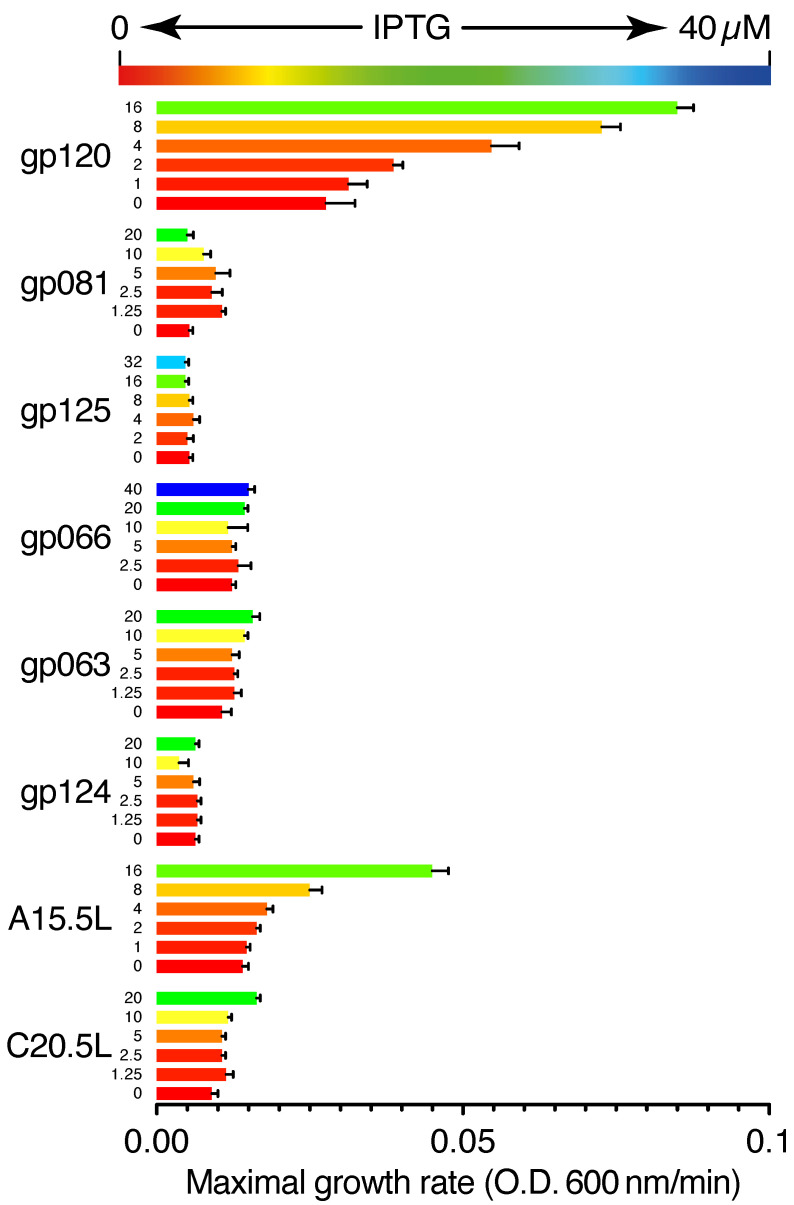
Positive genetic test for viroporin activity. The protein chimeras were expressed within a $K^{+}$-uptake deficient bacteria [35] to examine whether they trigger $K^{+}$ flux and increase the growth rate of bacteria.

**Figure 3 ijms-24-13828-f003:**
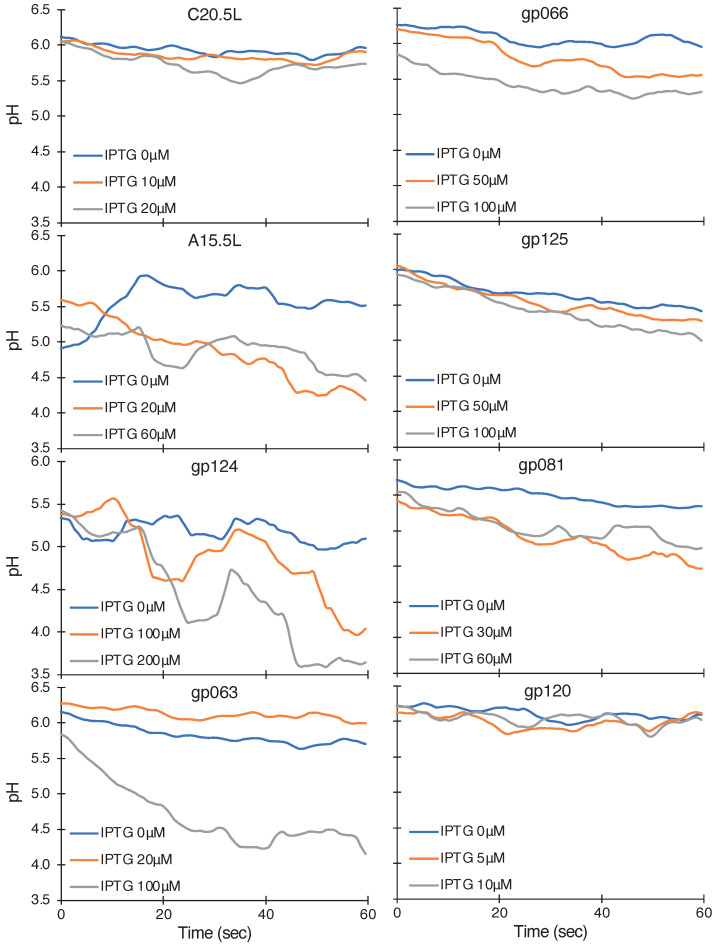
Fluorescent-based $H^{+}$ influx assay. The proton uptake was measured as function of fluorescence excitation peak maxima at 390 nm and 460 nm at several different protein induction levels governed by the IPTG concentration, as noted in the figure.

**Figure 4 ijms-24-13828-f004:**
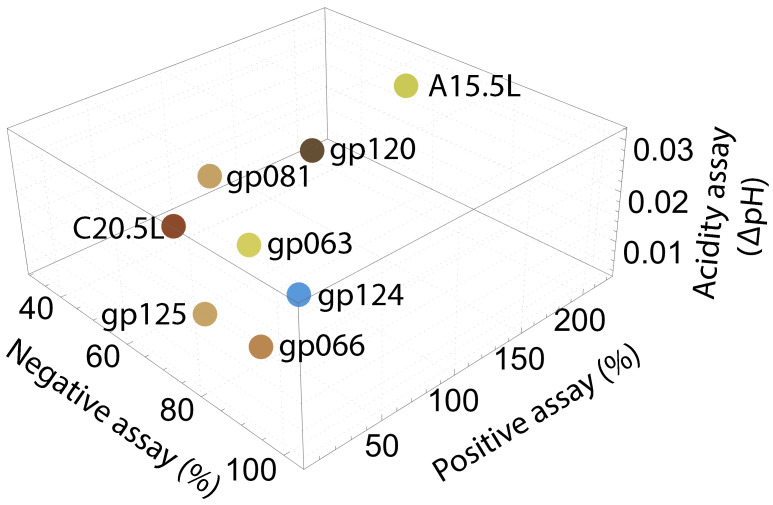
Diagram presenting a summary of the bacteria-based assay results for channel activity for the different viral proteins. The negative assay results represent the relative growth inhibition at 60 µM IPTG (Figure 1). The positive assay results are taken from Figure 2. The pH assay results are obtained from the slopes of linear regressions of the results presented in Figure 3.

**Figure 5 ijms-24-13828-f005:**
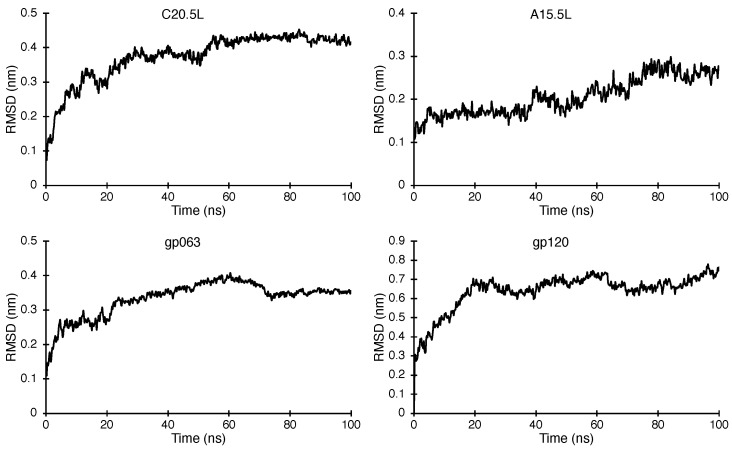
Backbone RMSD plots as a function of time for the molecular dynamics simulation of proteins: C20.5L, A15.5L, gp063, and gp120 within a pre-equilibrated 1,2-dipalmitoyl-sn-glycero-3-phosphocholine bilayer.

**Figure 6 ijms-24-13828-f006:**
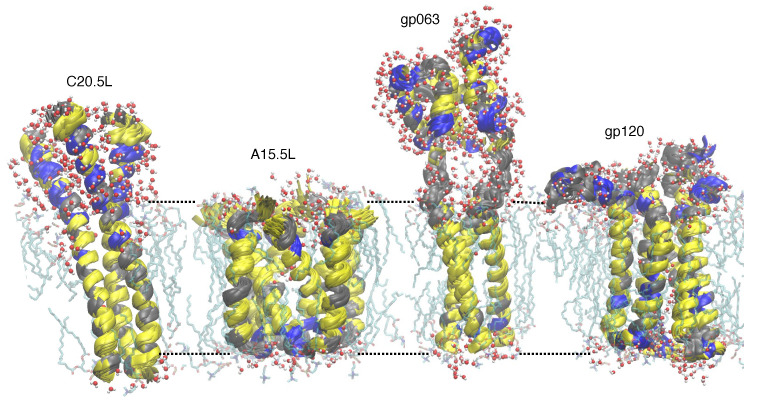
One hundred (100) ns MD simulated overlayed ensemble structures of C20.5L, A15.5L, gp063, and gp120 embedded within a triphasic lipid bilayer. Rough bilayer boundaries are shown in dotted lines with lipid and water molecules within 3 Å vicinity of proteins.

**Table 1 ijms-24-13828-t001:** List of transmembrane proteins chosen from MPXV genome with their accession numbers, length, and predicted transmembrane domains according to Phobius [30] and TMMHM [31].

Protein	Accession No.	Size	Predicted Transmembrane Domain
C20.5L	WBK62766.1	49	6–29 (according to Phobius)
			5–23 (according to TMMHM)
A15.5L	WBK62635.1	53	7–24 and 30–50 (according to Phobius)
			7–25 and 30–50 (according to TMMHM)
gp124	WBK62631.1	70	5–22 (according to Phobius)
			4–24 (according to TMMHM)
gp063	WBK62763.1	73	45–69 (according to Phobius)
			46–69 (according to TMMHM)
gp066	WBK62621.1	79	5–29 and 49–71 (according to Phobius)
			7–29 and 49–72 (according to TMMHM)
gp125	WBK62620.1	90	10–33 and 46–67 (according to Phobius)
			9–30 and 46–69 (according to TMMHM)
gp081	WBK62619.1	92	42–62 and 67–90 (according to Phobius)
			40–60 and 67–90 (according to TMMHM)
gp120	WBK62749.1	100	45–68 (according to Phobius)
			45–69 (according to TMMHM)

## Data Availability

All data are available upon request from the corresponding author.

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
