# Peer review of "Viroporins of Mpox Virus"

_ijms, 2023, doi:10.3390/ijms241813828_

Round 1

Reviewer 1 Report

Date: August 7, 2023

Title: Viroporins of Mpox virus

Authors: Basu et al.,

Critique:

Monkeypox virus is a large, enveloped DNA virus, classified in the genus Orthopaxvirus, family poxviridae with a genome size about 197kb. It is highly related to the smallpox, cowpox, and vaccinia viruses. One of the main characteristics of this group of viruses is to exclusively replicate in the cytoplasm of the infected cells.

The symptoms of monkeypox disease include blisters on the skin, fever, headache, and lymph node swelling. Treatments for monkeypox infection currently rely on the use of two FDA-approved drugs, Tecovirimat and Brincidofovir, each of which targets the extracellular viral maturation process and DNA polymerase respectively. Scientific community is also seeking to determine additional target sides on the monkeypox virus functional proteins including viroporins, because Influenza A virus M2 H+ channel was previously successfully targeted by aminoadamantanes to inhibit the virus.

In this study, investigators analyzed monkeypox virus transmembrane proteins with <100 aa in length for their potential to function as ion channels by employing three main techniques: Bacteria-based channel assay, AlphaFold2 and molecular dynamic simulation methods; and identified 8 potential proteins which can function as a viroporin. Specific inhibitors can be developed to inhibit their activity.

This paper is coming from a group of experts in the field and studies are well-designed and well-executed.

Specific Comments:

1.      Color designations on Fig. 1 are confusing. They do not represent a clear contrast. It can be improved perhaps by eliminating half of the IPTG concentration points and using sharp color differentials.

2.      On Fig. 3, there is a typing error for gp125, typed as gp125-1

3.      In Fig. 4 legend, there is a typing error “timer to hexamers”, should read as trimer to hexamers. In addition, annotations on Fig.4 panels are practically unreadable by a naked eye. These panels need to be revised such a way that annotations are readable.

4.      This manuscript can be further improved by including the references related to the viroporin activity of JC virus Agnoprotein as follows: 1. Suzuki et al., 2010, Plos Pathogens 6, e1000801; 2. Suzuki et al., 2013, PNAS 110: 18668-18673; 3. Pascale et al., 2014, J. Virol. 88: 6556-6575.

Author Response

See attached file that summarizes our response to the editor and both reviewers

Reviewer 2 Report

Problem: Monkeypox virus (MPXV) has emerged as a global threat. Few anti-viral drugs are available, and it is beneficial to characterize new drug targets in the virus.

Proposed solution: Channels are excellent drug targets, and we have identified viral ion channels for MPXV.

Findings: Bioinformatics analyses yielded eight transmembrane proteins smaller than 100 amino acids. Three independent bacteria-based assays have pointed to five of the eight proteins that exhibit ion channel activity.

Incomplete passage: Authors propose a tentative structure of four ion channels from their primary amino acid sequences, employing AlphaFold2 and molecular dynamic simulation methods. However, this passage lacks information about the molecular mechanisms of aquaporin activity as channels.

Author's claim: Despite the incomplete passage, the authors' claim that these results may represent the first steps in characterizing MPXV viroporins en route to developing blockers that inhibit their function still stands.

Reviewer's comments: The reviewer cannot agree to publish the manuscript unless the following amendments are made:

Table 1: Create and add figures showing the specific sequence and transmembrane domain of each ion channel, as well as potential gating mechanisms such as cation-pi interactions.

Figure 1, 2, and 3: Give statistics, such as R's Box plots.

Table 2: Remove. Make a diagram instead, such as a three-dimensional plot of three types of data to create clusters. Or perform statistical treatment such as reducing the dimension by principal component analysis. In common with the previous suggestion, the physical parameters of the helix region may be included using GRAVY and pI in ProtPram.

Figure 4-7: Remove.

Manuscript changes: The manuscript has been changed accordingly.

N/A

Author Response

(The authors gave the same response as above.)
